# Neighborhood Self-Dissimilarity Attention for Medical Image Segmentation

**Junren Chen[1], Rui Chen[2], Wei Wang[3], Junlong Cheng[1],**
**Gang Liang[4]\*, Lei Zhang[1]\*, Liangyin Chen[1]\***

[1] College of Computer Science, Sichuan University, Chengdu, China
[2] Department of Electronic Engineering, Tsinghua University, Beijing, China
[3] School of Automation, Chengdu University of Information Technology, Chengdu, China
[4] School of Cyber Science and Engineering, Sichuan University, Chengdu, China
`{lianggang, zhanglei, chenliangyin}@scu.edu.cn`

## Abstract

Medical image segmentation based on neural networks is pivotal in promoting digital health equity. The attention mechanism increasingly serves as a key component in modern neural networks, as it enables the network to focus on regions of interest, thus improving the segmentation accuracy in medical images. However, current attention mechanisms confront an accuracy-complexity trade-off paradox: accuracy gains demand higher computational costs, while reducing complexity sacrifices model accuracy. Such a contradiction inherently restricts the real-world deployment of attention mechanisms in resource-limited settings, thus exacerbating healthcare disparities. To overcome this dilemma, we propose a parameter-free Neighborhood Self-Dissimilarity Attention (NSDA), inspired by radiologists' diagnostic patterns of prioritizing regions exhibiting substantial differences during clinical image interpretation. Unlike pairwise-similarity-based self-attention mechanisms, NSDA constructs a size-adaptive local dissimilarity measure that quantifies element-neighborhood differences. By assigning higher attention weights to regions with larger feature differences, NSDA directs the neural network to focus on high-discrepancy regions, thus improving segmentation accuracy without adding trainable parameters directly related to computational complexity. The experimental results demonstrate the effectiveness and generalization of our method. This study presents a parameter-free attention paradigm, designed with clinical prior knowledge, to improve neural network performance for medical image analysis and contribute to digital health equity in low-resource settings. The code is available at https://github.com/ChenJunren-Lab/Neighborhood-Self-Dissimilarity-Attention.

## 1 Introduction

Medical image segmentation is pivotal in modern healthcare by enabling precise delineation of anatomical structures and radiological abnormalities within medical images [47]. This segmentation provides clinicians with valuable imaging information for personalized decision-making, thereby enhancing patient care and outcomes [11]. However, in resource-limited regions where expert-level radiological expertise is scarce, suboptimal medical image segmentation constitutes a significant barrier to equitable care [16], often leading to delayed or inaccurate diagnoses for complex conditions like multi-organ segmentation and tumor delineation [1, 45]. Such disparities create a diagnostic expertise gap between resource-rich and resource-poor healthcare systems, impeding equitable access to precision diagnostics [35]. To bridge this healthcare divide, neural networks have emerged as

---

\*Corresponding authors: Gang Liang, Lei Zhang and Liangyin Chen

39th Conference on Neural Information Processing Systems (NeurIPS 2025).

transformative solutions [12]. They are capable of automatically learning complex feature patterns from medical images [31], thereby overcoming the limitations of traditional manual methods, which are time-consuming and rely on clinical expertise. Therefore, neural networks can potentially promote digital health equity and become the dominant approach in medical image segmentation [4, 21].

Attention mechanisms [23], inspired by the human visual system [15, 34], are now essential for improving medical image segmentation in modern neural networks [67]. These attention mechanisms recalibrate the channel-wise or spatial importance within the feature map, enabling the network to selectively focus on the Regions of Interest (ROIs), thus improving the segmentation accuracy in medical images [49]. Modern attention mechanisms widely adopt advanced feature extraction strategies such as multi-branch architectures [41], multi-scale processing [52], large-kernel convolutions [42], and self-attention operators [25] to enhance accuracy in neural networks. However, such sophisticated designs inevitably increase model complexity, hindering practical deployment in low-resource settings [20], thereby compelling practitioners to forgo integrating these advanced attention mechanisms into their neural networks [61]. Although lightweight architectures reduce computational demands through compact convolution operators (e.g., depthwise separable convolutions) [7], group channel processing [52, 62], and sparse sampling strategies [39], such architectural simplifications often result in coarse-grained feature representations. This representational degradation can impair segmentation accuracy in medical images, as the accurate delineation of subtle radiological abnormalities and anatomical structures relies on fine-grained features [3, 43]. Consequently, the design of attention mechanisms confronts a fundamental architectural conundrum: the endeavor to improve segmentation accuracy runs counter to the maintenance of computational efficiency. This accuracy-complexity trade-off paradox exacerbates healthcare disparities by limiting equitable access to high-accuracy attention mechanisms for neural networks in resource-constrained environments.

To overcome the accuracy-complexity trade-off paradox, we propose a parameter-free Neighborhood Self-Dissimilarity Attention (NSDA) mechanism for neural networks. This approach focuses on element-neighborhood differences to boost segmentation accuracy without adding trainable parameters directly related to computational complexity. The proposed NSDA is inspired by two key principles in radiologists' image-reading practices [60]: (i) *Neighborhood Inspection* [59]. It represents a focal-to-contextual observational paradigm, wherein radiologists begin by focusing on a specific element (e.g., a pixel) or ROI in an image and then expand their attention to the neighborhood to synthesize contextual clues. (ii) *Difference Prioritization* [2]. Radiologists prioritize differences in medical images, as these features constitute essential biomarkers for clinical decision-making. Accordingly, we devise a dissimilarity measure based on complement of a Gaussian kernel to quantify the feature differences between each element and its neighborhood in the feature map. By assigning higher attention weights to regions with salient feature differences, this approach enhances segmentation accuracy. Notably, most existing difference-based approaches for medical image analysis operate in the multi-image paradigm [8, 18, 26, 40], relying on multiple feature images to characterize inter-image differences. Unlike these methods requiring inter-image analysis, NSDA processes local neighborhoods within single images, thereby avoiding multi-image comparisons yet retaining sensitivity to fine-grained differences. Furthermore, given the clinical prior knowledge that diagnostically critical patterns are often localized [19, 68], we introduce a Dynamic Neighborhood Scaling (DyNS) strategy for NSDA. DyNS adaptively regulates the neighborhood window size in NSDA across network hierarchies to prevent feature homogenization induced by aggregating long-range contextual information. Unlike pairwise-similarity-based self-attention mechanisms [25, 39, 57], the proposed NSDA focuses on element-neighborhood dissimilarity without adding parameters. Our radiologist-vision-inspired attention mechanism not only preserves original network complexity, but also demonstrates the potential to advance digital health equity in low-resource settings.

In summary, the main contributions of our study are as follows:

- Inspired by radiologists' neighborhood inspection and difference prioritization, we propose a parameter-free Neighborhood Self-Dissimilarity Attention (NSDA) that enables neural networks to focus on element-neighborhood differences, thereby boosting segmentation accuracy.

- We introduce a Dynamic Neighborhood Scaling (DyNS) strategy in NSDA, which adaptively adjusts NSDA's neighborhood window size across network hierarchies to prevent feature homogenization caused by aggregating excessive long-range information.

- Experimental results demonstrate that NSDA outperforms traditional attention mechanisms, highlighting their effectiveness and potential to promote digital health equity in low-resource settings.

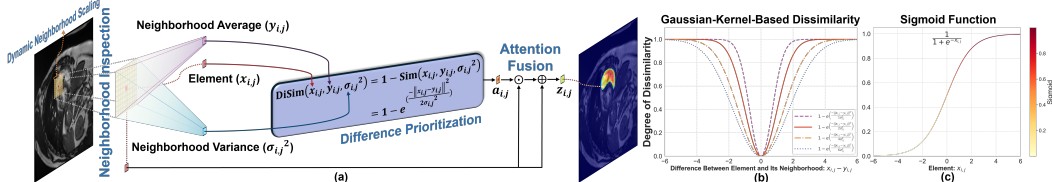

Figure 1: The Overview of our Neighborhood Self-Dissimilarity Attention (NSDA). (a) The proposed architecture of the NSDA illustrates how a Gaussian-kernel-based measure quantifies element-neighborhood dissimilarity, thereby assigning higher weights to elements with more salient differences. (b) The dissimilarity-driven weighting curve generated by NSDA. (c) The conventional sigmoid-based weighting curve employed in traditional attention mechanisms. Compared with traditional sigmoid-based methods, NSDA eliminates sign-induced bias, thereby ensuring equitably weighted contributions from features with opposing polarity.

## 2 Related Work

**Medical Image Segmentation** based on data-driven U-shaped Networks (U-Nets) [4, 53] has triggered a revolutionary trend in computer-aided diagnosis, demonstrating great potential to democratize access to precision medicine in resource-constrained clinical environments [12]. However, due to the inherent inductive bias of Convolutional Neural Networks (CNNs), U-Nets struggle to model long-range contextual relationships [64], limiting their ability to comprehensively characterize ROIs. To mitigate this limitation, subsequent U-Net variants expand receptive fields via parallel or cascaded modules [12, 52]. More recently, advanced approaches [10, 24, 32, 46] integrate Vision Transformers (ViTs) or Mamba architectures to model global information for enhanced contextual awareness. However, most existing methods still process all regions of an image equally during feature learning, failing to pay attention to diagnostically vital ROIs, and thus achieving limited segmentation accuracy. To bridge this gap, attention mechanisms have emerged as a pivotal solution [23], allowing improved segmentation accuracy by guiding networks to focus on salient regions in complex scenes [14, 54, 67].

**Visual Attention Mechanisms** in neural networks are broadly categorized into channel and spatial attention [23], recalibrating feature map weights to prioritize ROIs [49]. Channel attention mechanisms aggregate channel information via global pooling to generate compressed descriptor vectors [13, 50]. These vectors undergo nonlinear transformations through convolutions or Multi-Layer Perceptrons (MLPs) before yielding channel weights by a sigmoid function, enabling networks to prioritize semantically relevant targets [30]. However, these mechanisms inherently lack spatial perception capabilities [63]. Spatial attention mechanisms address this limitation by capturing positional dependencies between pixels or tokens [9, 29]. For instance, established self-attention architectures [25, 39] partition feature maps into tokens and compute pairwise similarity matrices to model global contextual relationships. While subsequent mechanisms primarily enhance feature representation based on channel or spatial dimensions [7, 52, 62], these methods rely on computationally intensive operations such as convolutions and MLPs, thus rendering them impractical for resource-limited settings and expanding the healthcare access divide [61]. Recent lightweight alternatives [55, 66] use simplified operations such as compact operators, partial computation, or sparse sampling to reduce complexity, yet often degrade fine-grained feature representation critical for medical image segmentation [12], leading to suboptimal accuracy gain for neural networks. Thus, overcoming the accuracy-complexity trade-off paradox is an urgent challenge in the research of attention mechanisms.

## 3 Methodology

**Overview**. We propose a radiologist-vision-inspired NSDA, which guides neural networks to focus on element-neighborhood differences to enhance segmentation accuracy while overcoming the accuracy-complexity trade-off paradox. Figure 1 illustrates a schematic of the proposed NSDA operating on individual elements of a feature map. First, we design a Dynamic Neighborhood Scaling to adaptively regulate NSDA's neighborhood boundaries across network layers, preventing feature homogenization caused by aggregating excessive long-range information. Second, we simulate a radiologist's neighborhood inspection by extending the analysis scope from each element to its

neighborhood, thereby aggregating contextual information. Third, we construct a dissimilarity measure that quantifies the dissimilarity between each element and its aggregated neighborhood representation, which in turn modulates attention weights based on the degree of dissimilarity, thus modeling the radiologist's difference prioritization. Finally, we perform Attention Fusion on the original element and attention weight to obtain the final output. Each element undergoes the aforementioned operations to generate attention-augmented feature maps, enabling element-wise discriminability critical for fine-grained medical image segmentation.

**Dynamic Neighborhood Scaling**. Traditional attention mechanisms capture the global context through global pooling or self-attention operations. However, in medical imaging, ROIs such as lesions or anatomical structures often occupy small spatial footprints, but this constraint is ignored by most existing approaches, leading to a suboptimal segmentation accuracy improvement for neural networks. Although straightforwardly constraining attention to a fixed-size window may initially seem beneficial, such a static window gradually exceeds the feature map size as the network depth increases, homogenizing critical local feature representations. To address this issue, we design a simple Dynamic Neighborhood Scaling (DyNS) strategy that adjusts the neighborhood window size in proportion to the resolution of the feature map (i.e., its spatial dimensions) at each network hierarchy. This process is formulated as follows:

$$b_L = \left\lfloor \frac{B_L}{S} \right\rfloor + c, \quad \{b_L, B_L, S\} \in \mathbb{N}_+, \quad c = \begin{cases} 1, & \text{if } \lfloor B_L/S \rfloor \text{ is even} \\ 2, & \text{otherwise} \end{cases}, \quad (1)$$

where $b_L$ and $B_L$ denote the neighborhood window size (width/height) and the corresponding feature map size in the $L$-th network layer, respectively. The scale factor $S$ is empirically set to 8. The constant $c$ ensures bilateral symmetry by ensuring that $b_L$ is an odd integer, which establishes equidistant left-right boundaries around target elements to facilitate centered neighborhood aggregation. As a result, DyNS preserves fine-grained features while mitigating the feature homogenization arising from the aggregation of extraneous global contexts, which is crucial for segmenting small ROIs.

**Neighborhood Inspection**. In medical image analysis, the semantic interpretation of individual elements (e.g., pixels) is primarily determined by localized contextual patterns rather than long-range dependencies. Although natural images benefit from global region interactions for holistic scene understanding, medical imaging exhibits a distinct behavior in which excessive reliance on long-range dependencies introduces feature homogenization. Inspired by radiologists' neighborhood inspection (i.e., the focal-to-contextual observational paradigm), we model per-element neighborhood contextual dependencies in feature maps. This design simulates clinical reasoning by prioritizing fine-grained patterns within constrained anatomical regions, effectively avoiding feature homogenization while retaining discriminative details essential for segmentation. Specifically, for a given target element $x_{i,j}$, we aggregate features from its centered $b_L \times b_L$ window. As shown in Figure 1, this process generates two statistical metrics: an average $y_{i,j}$ encoding the contextual representation of the neighborhood window [69] and a variance $\sigma^2_{i,j}$ quantifying the feature variability within the localized neighborhood [33]. This process is expressed as follows:

$$y_{i,j} = \frac{1}{b_L \times b_L} \sum_{n=i-l}^{i+l} \sum_{m=j-l}^{j+l} (x_{m,n}), \sigma^2_{i,j} = \frac{1}{b_L \times b_L} \sum_{n=i-l}^{i+l} \sum_{m=j-l}^{j+l} (x_{m,n} - y_{i,j})^2, l = \frac{b_L - 1}{2}, \quad (2)$$

where $i, j \in \mathbb{N}_+$ denote the 2D spatial coordinates in the feature map and $l \in \mathbb{N}_+$ defines the minimum distance from the element $x_{i,j}$ to the boundaries of its local neighborhood window.

**Difference Prioritization**. The dissimilarity between an element and its neighborhood depicts the presence of complex high-frequency patterns, such as textural variations, within localized regions. This observation implies that quantifying discrepancies, as opposed to similarities, offers heightened sensitivity in detecting subtle anatomical boundaries and radiological anomalies. Motivated by radiologists' image-reading practice of prioritizing discrepancies, we propose a novel dissimilarity measure that adaptively assigns higher attention weights to elements exhibiting more salient deviations from their neighborhood context. Although the Euclidean distance is widely used as a dissimilarity measure, it violates two fundamental requirements for attention mechanisms [23, 49, 67]: (i) a nonlinear response to feature variations and (ii) a bounded output within $[0, 1]$. In contrast, the Gaussian kernel [6, 36] satisfies both requirements, yet intrinsically captures similarity, not dissimilarity. To address this issue, we derive our dissimilarity measure by the complement of the Gaussian kernel. Specifically, we first compute the Gaussian-kernel-based similarity score between the element $x_{i,j}$

and its contextually aggregated neighborhood representation $y_{i,j}$. Here, $y_{i,j}$ is obtained by the local average pooling [13, 69]. We then derive a dissimilarity score, exploiting the complementarity constraint between similarity and dissimilarity scores, which sum to 1. The above process generates the corresponding attention weight $a_{i,j} \in [0, 1]$ for element $x_{i,j}$ as follows:

$$a_{i,j} = \text{DiSim}(x_{i,j}, y_{i,j}, \sigma_{i,j}^2) = 1 - \text{Sim}(x_{i,j}, y_{i,j}, \sigma_{i,j}^2) = 1 - e^{\left(\frac{-\|x_{i,j} - y_{i,j}\|^2}{2\sigma_{i,j}^2}\right)}, \quad (3)$$

where $y_{i,j}$ and $\sigma_{i,j}^2$ are derived directly from Equation 2. The function $\text{Sim}(\cdot) \in [0, 1]$ represents the Gaussian kernel, which quantifies the similarity between $x_{i,j}$ and $y_{i,j}$ [6]. The proposed Gaussian-kernel-based dissimilarity measure ensures adherence to the boundary conditions of the attention score while strictly enforcing the complementarity constraint of similarity.

**Attention Fusion**. The attention mechanism is typically integrated into neural networks by element-wise multiplication between weight matrices $A \in \mathbb{R}^{H \times W}$ and feature maps $X \in \mathbb{R}^{H \times W}$. However, in this conventional fusion method, the weight $a_{i,j} \in A$ falls in the range $[0, 1]$, resulting in $x_{i,j} \cdot a_{i,j} \leq x_{i,j}$, which diminishes the feature representation. To address this limitation, inspired by [23, 27, 49], we incorporate a residual connection for the attention fusion. For a target element $x_{i,j}$ with its associated attention weight $a_{i,j}$, the attention-augmented feature is yielded via:

$$z_{i,j} = x_{i,j} \cdot a_{i,j} + x_{i,j}. \quad (4)$$

The proposed method performs element-wise attention fusion per feature map. Given an input feature map $X \in \mathbb{R}^{H \times W}$ and its corresponding attention weight matrix $A \in \mathbb{R}^{H \times W}$, according to Equation 4, the final output feature map $Z \in \mathbb{R}^{H \times W}$ is computed through:

$$Z = (X \odot A) \oplus X, \quad X = \begin{pmatrix} x_{1,1} & \cdots & x_{1,W} \\ \vdots & \ddots & \vdots \\ x_{H,1} & \cdots & x_{H,W} \end{pmatrix}, \quad A = \begin{pmatrix} a_{1,1} & \cdots & a_{1,W} \\ \vdots & \ddots & \vdots \\ a_{H,1} & \cdots & a_{H,W} \end{pmatrix}, \quad (5)$$

where $\odot$ and $\oplus$ denote element-wise multiplication and element-wise addition, respectively. Our method eliminates learnable parameters (e.g., convolutional layers or MLPs) directly related to computational complexity, thus preserving its compatibility in resource-limited deployment scenarios.

## 4 Experiments

**Objective**. We perform experiments to study the ability of NSDA, designed to address the following research questions (RQs):

- **RQ1**: How does our Neighborhood Self-Dissimilarity Attention (NSDA) behave in the neural networks for medical image segmentation, compared with the traditional attention mechanisms?
- **RQ2**: Does the inference mechanism of NSDA possess explainability?
- **RQ3**: How is the generalizability of NSDA across heterogeneous architectures and tasks?
- **RQ4**: How does the neighborhood window size in NSDA affect segmentation accuracy, and is Dynamic Neighborhood Scaling (DyNS) effective?
- **RQ5**: How sensitive is the scale factor in DyNS to segmentation accuracy?
- **RQ6**: How does the variance coefficient in NSDA's Gaussian kernel affect segmentation accuracy?
- **RQ7**: Can dissimilarity in NSDA be quantified using other measures, like the Euclidean distance?
- **RQ8**: Which is more critical for medical image segmentation in NSDA, similarity or dissimilarity?

**Datasets**. We evaluate the effectiveness of NSDA on three prominent medical image segmentation benchmarks with diverse modalities and scales: Synapse (multi-organ abdominal CT) [45], ACDC (cardiac MRI) [5], and BUSI (breast ultrasound) [1]. To thoroughly assess generalization capabilities for the proposed NSDA, we extend validation to COVID-19 pneumonia lesion detection (CPLDet) [51], endoscopic bladder tissue classification (EBTCls) [38], and natural image segmentation (VOCSeg, combining PASCAL VOC07 [17] and VOC12 [65]).

**Baselines**. To conduct a comprehensive comparison, we not only compare the well-known attention mechanisms in medical image tasks like SE [30], CBAM [63], and MSCAM [52], but also compare

Table 1: A quantitative comparison of the proposed NSDA and other baselines integrated into segmentation networks (U-Net TransUNet, UNeXt, and TinyU-Net) across datasets (Synapse, ACDC, and BUSI), using mean DSC (%) for main evaluation metric. The highest score is marked in **bold**.

| Network | Params (M) | FLOPs (G) | Tput (FPS) | Synapse | ACDC | BUSI | Network | Params (M) | FLOPs (G) | Tput (FPS) | Synapse | ACDC | BUSI |
|---|---|---|---|---|---|---|---|---|---|---|---|---|---|
| U-Net [53] | 19.5820 | 101.7706 | 298.34 | 78.18 | 91.77 | 71.60 | TransUNet [10] | 93.2317 | 64.6085 | 68.26 | 79.89 | 92.49 | 72.70 |
| + SE [30] | 19.7018 | 101.8026 | 246.19 | 75.44 | 90.53 | 71.90 | + SE [30] | 93.4145 | 64.6182 | 64.55 | 79.46 | 92.42 | 73.71 |
| + CBAM [63] | 19.7052 | 101.8370 | 117.49 | 78.06 | 90.85 | 66.29 | + CBAM [63] | 93.4176 | 64.6367 | 53.32 | 79.23 | 92.41 | 72.48 |
| + ECA [61] | 19.5820 | 101.8023 | 251.74 | 76.48 | 90.60 | 69.89 | + ECA [61] | 93.2317 | 64.6179 | 64.70 | 79.45 | 92.48 | 68.15 |
| + BAM [50] | 19.8954 | 102.6914 | 150.34 | 77.48 | 92.03 | 71.23 | + BAM [50] | 93.7061 | 65.0218 | 52.75 | 79.04 | 92.46 | 72.94 |
| + SimAM [66] | 19.5820 | 101.7706 | 242.82 | 77.29 | 91.68 | 70.97 | + SimAM [66] | 93.2317 | 64.6085 | 64.40 | 79.76 | 92.35 | 67.63 |
| + CA [29] | 19.6823 | 101.8463 | 166.37 | 76.00 | 90.04 | 70.34 | + CA [29] | 93.3766 | 64.6368 | 59.14 | 79.62 | 91.64 | 31.52 |
| + GC [9] | 20.0679 | 101.8043 | 210.49 | 77.50 | 91.38 | 71.47 | + GC [9] | 93.9692 | 64.6199 | 60.97 | 82.90 | 92.42 | 73.89 |
| + MHSA [25] | 23.4255 | 119.4873 | 183.32 | 78.69 | 91.56 | 71.59 | + MHSA [25] | 99.0908 | 72.7292 | 59.22 | 81.55 | 92.48 | 71.12 |
| + MWSA [39] | 19.8966 | 105.8307 | 91.26 | 43.28 | 72.73 | 55.25 | + MWSA [39] | 98.7160 | 64.6284 | 58.46 | 74.76 | 92.13 | 68.56 |
| + MLKA [62] | 23.2917 | 121.6817 | 132.89 | 79.22 | 92.35 | 71.54 | + MLKA [62] | 98.6799 | 73.0168 | 54.59 | 78.65 | 92.46 | 73.38 |
| + CGA [13] | 20.0660 | 104.9455 | 159.53 | 78.01 | 90.45 | 70.53 | + CGA [13] | 93.8239 | 65.5551 | 56.72 | 80.68 | 92.46 | 71.96 |
| + CAA [7] | 21.5718 | 111.6123 | 183.90 | 77.72 | 90.79 | 71.03 | + CAA [7] | 96.2244 | 68.9593 | 60.50 | 80.46 | 92.42 | 71.27 |
| + CMO [28] | 22.4890 | 115.3436 | 183.36 | 77.17 | 92.15 | 56.41 | + CMO [28] | 97.6486 | 70.7833 | 60.17 | 79.57 | 92.29 | 72.28 |
| + MSCAM [52] | 23.7311 | 122.9160 | 85.54 | 76.22 | 91.40 | 70.34 | + MSCAM [52] | 99.4458 | 73.7508 | 47.79 | 19.03 | 86.03 | 71.07 |
| + NSDA (Ours) | 19.5820 | 101.7706 | 130.81 | **81.62** | **92.56** | **74.46** | + NSDA (Ours) | 93.2317 | 64.6085 | 63.51 | **83.81** | **92.77** | **75.29** |
| UNeXt [58] | 1.4721 | 1.1628 | 238.36 | 72.40 | 89.18 | 63.71 | TinyU-Net [12] | 0.4816 | 3.3855 | 172.42 | 78.33 | 91.68 | 72.24 |
| + SE [30] | 1.4743 | 1.1668 | 203.38 | 71.87 | 88.74 | 60.38 | + SE [30] | 0.5687 | 3.4171 | 141.93 | 77.48 | 91.56 | 71.24 |
| + CBAM [63] | 1.4747 | 1.1878 | 130.76 | 71.64 | 87.97 | 40.24 | + CBAM [63] | 0.5715 | 3.4514 | 100.90 | 79.23 | 90.14 | 70.82 |
| + ECA [61] | 1.4721 | 1.1668 | 205.20 | 71.57 | 88.67 | 62.02 | + ECA [61] | 0.4816 | 3.4169 | 143.44 | 79.53 | 91.94 | 69.92 |
| + BAM [50] | 1.4782 | 1.1894 | 149.58 | 73.17 | 89.11 | 65.47 | + BAM [50] | 0.7097 | 4.2795 | 102.13 | 79.00 | 91.66 | 71.65 |
| + SimAM [66] | 1.4721 | 1.1628 | 209.87 | 70.78 | 88.15 | 64.79 | + SimAM [66] | 0.4816 | 3.3855 | 145.11 | 78.03 | 91.47 | 70.29 |
| + CA [29] | 1.4767 | 1.1719 | 157.17 | 26.96 | 69.59 | 20.15 | + CA [29] | 0.5564 | 3.4595 | 113.92 | 80.05 | 91.67 | 71.91 |
| + GC [9] | 5.7934 | 1.1763 | 180.54 | 72.32 | 89.17 | 65.18 | + GC [9] | 0.8351 | 3.4187 | 130.87 | 78.31 | 92.00 | 71.57 |
| + MHSA [25] | 1.5434 | 1.5655 | 168.57 | 72.29 | 89.04 | 63.45 | + MHSA [25] | 3.2746 | 20.5653 | 122.07 | 80.02 | 92.07 | 71.23 |
| + MWSA [39] | 1.4950 | 1.6661 | 99.51 | 58.38 | 60.75 | 16.80 | + MWSA [39] | 0.7301 | 7.4120 | 74.89 | 73.24 | 90.53 | 66.35 |
| + MLKA [62] | 1.5596 | 2.0617 | 143.39 | 72.35 | 88.23 | 61.43 | + MLKA [62] | 3.1991 | 22.7904 | 98.55 | 80.29 | 92.11 | 71.89 |
| + CGA [13] | 1.4944 | 1.5732 | 153.03 | 73.03 | 89.44 | 60.28 | + CGA [13] | 0.8487 | 6.5342 | 109.02 | 79.21 | 92.22 | 72.20 |
| + CAA [7] | 1.5127 | 1.4860 | 184.00 | 51.63 | 79.25 | 51.25 | + CAA [7] | 1.9319 | 12.9506 | 125.17 | 80.37 | 92.37 | 71.34 |
| + CMO [28] | 1.5273 | 1.5002 | 173.81 | 50.81 | 86.62 | 28.80 | + CMO [28] | 2.5955 | 16.5535 | 122.98 | 79.58 | 92.34 | 71.46 |
| + MSCAM [52] | 1.5593 | 2.0073 | 106.12 | 11.29 | 10.01 | 20.68 | + MSCAM [52] | 3.5083 | 23.9657 | 75.90 | 37.84 | 79.83 | 71.89 |
| + NSDA (Ours) | 1.4721 | 1.1628 | 183.98 | **77.39** | **90.18** | **67.60** | + NSDA (Ours) | 0.4816 | 3.3855 | 126.29 | **81.57** | **92.73** | **75.09** |

the outstanding attention mechanisms in natural image/remote sensing image tasks such as CA [29], MLKA [62], CAA [7], and pairwise-similarity-based self-attention mechanisms like MHSA [25], MWSA [39]. To compare attention mechanisms that can be deployed in resource-limited settings, we incorporate lightweight variants (e.g., ECA [61], SimAM [66]) for baseline comparison.

**Implementation Details**. Attention mechanisms are typically integrated into the final layer of feature extraction modules in established medical image segmentation architectures (U-Net [53], TransUNet [10], UNeXt [58], and TinyU-Net [12]) following conventional integration paradigms [49]. This conventional integration scheme preserves the architectural integrity of the backbone networks. We conduct experiments on an NVIDIA GeForce RTX 4090 GPU using the PyTorch framework. The models are trained for 300 epochs using the Adam optimizer [37], with a composite loss function that combines cross-entropy and Dice loss [12]. The initial learning rate is set to $1 \times 10^{-4}$ and decayed using a cosine annealing scheduler, with a minimum value of $1 \times 10^{-6}$. Synapse and ACDC replicate TransUNet's setup [44, 45], while EBTCls adopts the method from [38] for processing images. We adopt different splits for each benchmark: a 6:2:2 ratio (train/validation/test) for BUSI, an 8:1:1 ratio for NCPDet, and a 9:1 train-test split for VOCSeg. All input images are resized to $256 \times 256$ resolution. Following the training strategy proposed in [22], data augmentation (flip, rotation) is performed for the first 270 epochs and turned off for the last 10% of training.

**Evaluation Metrics**. To evaluate model performance, we use the following evaluation metrics: the Dice Similarity Coefficient (DSC) for segmentation tasks [12], the mean Average Precision (mAP) for detection tasks [22], and the Top-1 Accuracy (Top-1 Acc), the mean Recall (mRec), the mean Precision (mPrec) for classification tasks. The computational efficiency of the model is assessed using the number of Parameters (Params), Floating-Point Operations (FLOPs), and Throughput (Tput) to quantify model size, computational complexity, and inference speed, respectively. The reported FLOPs correspond to twice the Multiply-Accumulate Operations (MACs) measured by THOP [22].

## 5 Results and Discussion

**Quantitative Analysis (RQ1)**. Table 1 presents a comprehensive comparison of established neural network architectures integrated with distinct attention mechanisms in three benchmark datasets for medical image segmentation. From these results, we have the following observations:

- **NSDA achieves the most significant performance gains** for neural network architectures across medical image segmentation benchmarks. The experimental results demonstrate that our method achieves consistent state-of-the-art performance improvements integrated into various segmentation networks (U-Net, TransUNet, UNeXt, and TinyU-Net) across multiple benchmarks (Synapse, ACDC, and BUSI). Specifically, when integrated into these network architectures, NSDA consistently achieves higher mean DSC (mDSC). For instance, the NSDA-augmented UNeXt achieves substantial gains of +4.99%, 1.00%, and +3.89% in mean DSC over the original UNeXt on the Synapse, ACDC, and BUSI datasets, respectively, significantly outperforming other attention baselines. The observed performance gains stem from NSDA's element-wise neighborhood modeling. It prevents feature homogenization through size-adaptive context aggregation and fine-grained feature representations. Our NSDA guides the network's attention toward regions with salient differences, thus enhancing its ability to detect complex anatomical structures and subtle radiological anomalies.

- **Traditional attention mechanisms demonstrate limited efficacy** in optimizing neural networks for medical image segmentation. As evidenced by Table 1, 168 baseline resutls of attention-integrated architectures reveal that 79% (132 out of 168) underperformed their original networks in medical image segmentation tasks. Notably, TransUNet achieves superior segmentation accuracy despite containing 91M more parameters than the lightweight UNeXt (Table 1). However, most attention modules with parameter counts below this threshold degrade segmentation performance, exposing fundamental flaws in conventional attention architectures rather than overfitting. These methods depend on global computation operators (e.g., pooling, self-attention), which induce excessive feature homogenization during contextual aggregation, progressively eroding anatomically critical patterns crucial for precise delineation. Unlike traditional attention mechanisms, our approach addresses the limitation of coarse-grained feature representation by explicitly modeling element-neighborhood dissimilarity to capture fine-grained differences. This strategy effectively guides the network's focus toward regions of interest (ROIs), enhancing segmentation accuracy.

- **NSDA has the potential to promote digital health equity** in resource-limited settings. Compared with traditional attention mechanisms and lightweight variants, our NSDA overcomes the long-standing accuracy-complexity trade-off paradox, achieving optimal performance in segmentation accuracy, parameter count, computational complexity (Table 1). For example, the NSDA-augmented TinyU-Net preserves the architectural compactness of the original model (0.48M parameters, 3.39G FLOPs) yet elevates segmentation accuracy by +3.24% mean DSC with a slight inference speed reduction on the Synapse dataset. Remarkably, NSDA-equipped TinyU-Net exhibits 99.49% fewer parameters and 94.75% lower computational complexity than TransUNet (0.48M vs. 93.23M Params; 3.39G vs. 64.61G FLOPs), while delivering $1.85\times$ faster inference speed (126.29 vs. 68.26 FPS) and superior mean DSC scores on Synapse (+1.68%), ACDC (+0.24%), and BUSI (+2.39%). This efficiency stems from NSDA's parameter-free design, which eschews resource-intensive operations such as convolutions or multi-layer perceptrons. Instead, it employs neighborhood aggregation based on PyTorch framework alongside a Gaussian-kernel-based dissimilarity measure to focus on fine-grained differences vital for medical image analysis. Such parameter-free architecture enables seamless integration into diverse backbones of neural networks and empowers lightweight models (e.g., UNeXt, TinyU-Net) to approximately rival the accuracy of resource-intensive models (e.g., TransUNet) — a critical advancement toward digital health equity in resource-limited settings.

**Qualitative Analysis (RQ2)**. To examine the explainability of attention mechanisms, we utilize Grad-CAM [56] to visualize the inference mechanism of these attention-integrated neural networks. Figure 2 presents comparative qualitative results of various attention-integrated U-Net variants on three medical imaging benchmarks. Grad-CAM visualizations reveal that our NSDA improves U-Net's ability to segment complex ROIs (e.g., the pancreas, right ventricle, and benign breast tumors), which the baseline U-Net fails to segment adequately. Qualitative segmentation results demonstrate that NSDA-augmented U-Net achieves significantly improved agreement with ground truth labels. These results highlight **NSDA's outstanding explainability** in medical image segmentation. This high explainability of NSDA can be attributed to the explicit embedding of radiologists' prior knowledge (*Neighborhood Inspection* and *Difference Prioritization*) into NSDA's architectural design.

**Generalization Evaluation (RQ3)**. We substitute the original attention module in the attention-augmented architecture (AttU-Net [48]) with NSDA to further assess its architectural generalization as a drop-in replacement. We find that NSDA reduces AttU-Net's computational complexity while enhancing segmentation accuracy compared with its original attention module (Table 2). Moreover, NSDA achieves consistent accuracy gains across various network architectures, demonstrating its

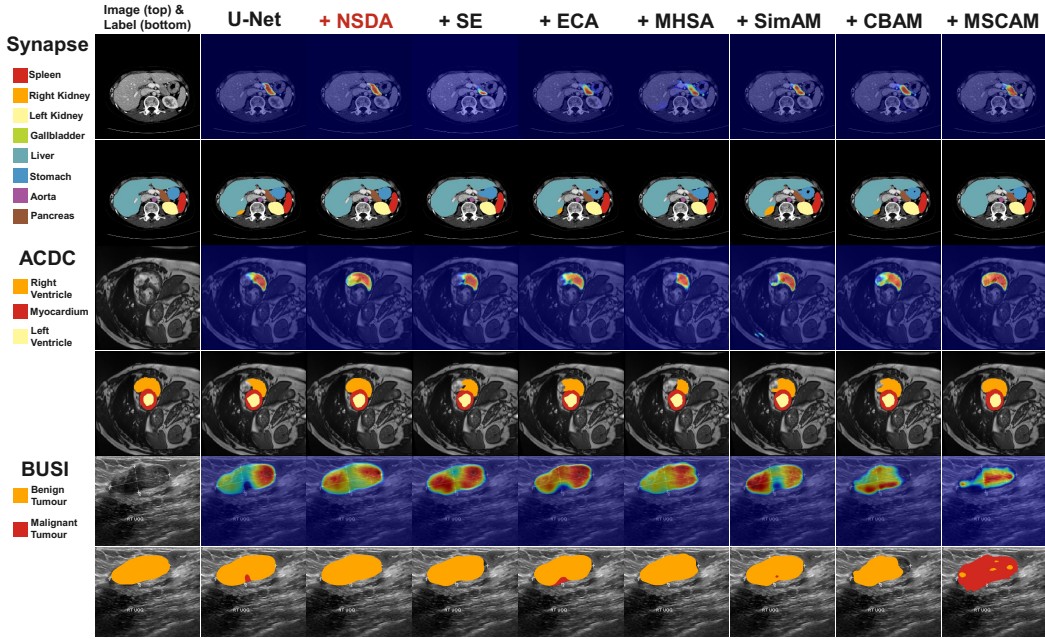

Figure 2: Qualitative results across various benchmarks. Odd rows: Grad-CAM [56] visualizations of attention-integrated U-Nets for segmenting the pancreas (Synapse [45]), right ventricle (ACDC [5]), and benign tumors (BUSI [1]). Warmer colors (e.g., red) indicate higher attention weights. Even rows: segmentation outcomes of attention-integrated U-Nets.

Table 2: Generalization performance of attention mechanisms across medical image tasks and a natural image segmentation task. OrigAttn denotes the original attention block in AttU-Net. Attention roles: '+' (complement), '←' (substitute). The highest score is marked in **bold**.

| Synapse (Medical Image Segmentation) | | | | CPLDet (Medical Image Detection) | | | | EBTCls (Medical Image Classification) | | | | VOCSeg (Natural Image Segmentation) | | |
|---|---|---|---|---|---|---|---|---|---|---|---|---|---|---|
| Network | Params | FLOPs | mDSC | Network | mAP50 | mAP75 | mAP | Network | Top-1 Acc | mRec | mPrec | Network | mIoU | mPA |
| AttU-Net [48] | 34.87 | 133.01 | 78.96 | YOLOX [22] | 81.65 | 53.02 | 49.68 | ResNet34 [27] | 60.93 | 53.52 | 46.91 | U-Net [53] | 45.46 | 57.69 |
| OrigAttn←SE [30] | 34.56 | 130.81 | 77.61 | + SE [30] | 81.47 | 52.68 | 49.26 | + SE [30] | 56.64 | 53.46 | 49.22 | + SE [30] | 41.09 | 52.14 |
| OrigAttn←ECA [61] | 34.51 | 130.81 | 77.38 | + ECA [61] | 79.31 | 50.66 | 47.42 | + ECA [61] | 56.61 | 48.58 | 42.24 | + ECA [61] | 40.92 | 51.58 |
| OrigAttn←MHSA [25] | 35.91 | 139.38 | 78.44 | + MHSA [25] | 80.67 | 53.03 | 47.41 | + MHSA [25] | 56.08 | 50.02 | 44.89 | + MHSA [25] | 28.00 | 37.12 |
| OrigAttn←SimAM [66] | 34.51 | 130.79 | 78.48 | + SimAM [66] | 79.01 | 52.88 | 48.93 | + SimAM [66] | 62.29 | 53.85 | 46.78 | + SimAM [66] | 48.06 | 59.45 |
| OrigAttn←CBAM [63] | 34.56 | 130.83 | 78.24 | + CBAM [63] | 80.56 | 55.17 | 50.45 | + CBAM [63] | 54.48 | 51.67 | 47.79 | + CBAM [63] | 34.53 | 44.56 |
| OrigAttn←MSCAM [52] | 36.03 | 141.08 | 73.56 | + MSCAM [52] | 53.84 | 10.62 | 20.94 | + MSCAM [52] | 52.37 | 48.41 | 45.92 | + MSCAM [52] | 38.47 | 51.33 |
| OrigAttn←NSDA (Ours) | 34.51 | 130.79 | **79.88** | + NSDA (Ours) | **82.04** | **55.83** | **50.92** | + NSDA (Ours) | **62.43** | **54.12** | **70.80** | + NSDA (Ours) | **48.55** | **61.04** |

cross-architecture generalization (Table 1). To further evaluate NSDA's cross-task generalization capacity, we integrate the attention mechanism into three network architectures: U-Net [53] for natural image segmentation, YOLOX [22] for medical image detection, and ResNet34 [27] for medical image classification. As shown in Table 2, NSDA consistently outperforms established attention baselines (e.g., CBAM [63], MHSA [25]) across tasks. These results validate the **generalization of NSDA** across architectures and tasks. The superior generalization of NSDA can be attributed to its dynamic neighborhood feature representation, which adaptively identifies input-dependent discrepancies between elements and their local context, rather than relying on static weights.

**Ablation Analysis (RQ4)**. To investigate the role of DyNS in NSDA, we replace DyNS with static neighborhood window sizes ($K \in \mathbb{N}_+$) in all network hierarchies. Since its complete removal leaves NSDA's neighborhood window size $b_L$ (Equation 1) undefined, $K \in \mathbb{N}_+$ must be manually set as a static neighborhood window size. As shown in Figure 3, the segmentation accuracy difference ($\Delta$mDSC) between NSDA-augmented networks with and without DyNS exhibits a bell-shaped trend on the multi-scale medical segmentation benchmark (Synapse). The $\Delta$mDSC initially increases with the static neighborhood window size $K$, peaks at $K = (31, 31)$, and subsequently declines with further increases in $K$. In addition, we find that aggregating features from a global region ($K = (H, W)$) degrades the segmentation accuracy. These results reveal two fundamental limitations in contextual modeling for medical image segmentation: (i) overly small-scale localized context aggregation fails to capture holistic structural patterns in ROIs, and (ii) excessively large-scale context

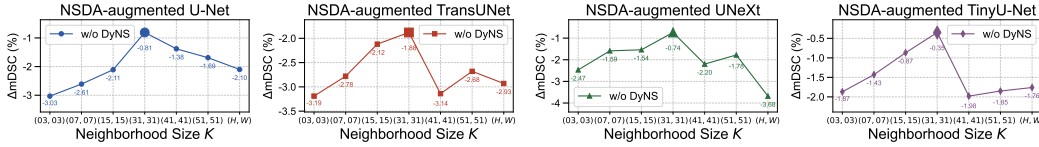

Figure 3: The results ($\Delta$mDSC (%)) of ablation analysis for DyNS on Synapse dataset. $H \in \mathbb{N}_+$ and $W \in \mathbb{N}_+$ in $K$ denote the height and width of the feature map, respectively. Each data point represents the segmentation accuracy difference ($\Delta$mDSC) between DyNS-free NSDA-augmented network architecture and its DyNS-equipped counterpart.

Table 3: The results (mDSC (%)) of sensitivity analysis for the scale factor $S$ (Equation 1) and the variance coefficient of Gaussian kernel (Equation 3) in NSDA on Synapse dataset. We report the mean and standard deviation ($mean_{std}$) from three independent runs with different random seeds. $\spadesuit$ and $\heartsuit$ denote U-Net and TinyU-Net with NSDA, respectively. The highest score is marked in **bold**.

| Network | Scale Factor in DyNS | | | | | Variance Coefficient of NSDA's Gaussian Kernel | | | |
|---|---|---|---|---|---|---|---|---|---|
| | $S = 2$ | $S = 4$ | $S = 8$ (Ours) | $S = 16$ | $S = 32$ | $\sigma_{i,j}^2$ | $2\sigma_{i,j}^2$ (Ours) | $4\sigma_{i,j}^2$ | $6\sigma_{i,j}^2$ |
| $\spadesuit$ | $80.68_{0.80}$ | $79.96_{0.55}$ | $\mathbf{81.35_{0.52}}$ | $79.74_{0.64}$ | $78.94_{0.83}$ | $80.49_{0.56}$ | $\mathbf{81.35_{0.52}}$ | $78.98_{0.66}$ | $78.74_{0.59}$ |
| $\heartsuit$ | $80.54_{0.69}$ | $79.37_{0.82}$ | $\mathbf{81.09_{0.46}}$ | $79.51_{0.74}$ | $80.29_{0.55}$ | $80.23_{0.37}$ | $\mathbf{81.09_{0.46}}$ | $79.88_{0.48}$ | $80.01_{0.41}$ |

aggregation causes feature homogenization, which diminishes anatomically discriminative patterns. Therefore, both excessively large window sizes and overly small window sizes lead to suboptimal accuracy due to their static nature, demonstrating the **necessity of balanced context aggregation**. Notably, as shown in Figure 3, all NSDA-augmented networks using the static neighborhood window size instead of the proposed DyNS exhibit a consistent accuracy degradation (negative $\Delta$mDSC) compared with DyNS-equipped NSDA-augmented networks, empirically confirming the **effectiveness of DyNS** in our NSDA. These results stem from DyNS dynamically adjusting the neighborhood window size based on the resolution of the feature map at each network hierarchy, which mitigates feature homogenization caused by aggregating overextended contextual ranges.

**Sensitivity Analysis (RQ5 & RQ6)**. We evaluate the individual effects of the DyNS scale factor $S$ (Equation 1) and the variance coefficient (Equation 3) for NSDA on segmentation accuracy by systematically varying their values (Table 3). We observe that NSDA-augmented networks achieve optimal segmentation accuracy with a scale factor of $S = 8$. The results in Table 3 further confirm the **importance of the scale factor trade-off**, which directly controls the size of the neighborhood window (Equation 1). We attribute this to a trade-off in neighborhood window size: an undersized neighborhood (large $S$) fails to capture essential context, while an oversized neighborhood (small $S$) leads to the dilution of discriminative local features. In addition, we find that NSDA-augmented networks achieve optimal segmentation accuracy with a Gaussian kernel variance coefficient of 2 (Table 3), empirically supporting the **effectiveness of this moderately low variance coefficient**. Our NSDA's variance coefficient of the Gaussian kernel aligns with the default setting in classical machine learning methods, such as SVM and PCA [6, 36]. The underlying mechanism is illustrated by the Gaussian kernel's weight profile in Figure 1(b). A larger variance coefficient produces a broader, flatter curve that dampens sensitivity to fine-grained differences, whereas a smaller coefficient yields a sharper, more localized curve that amplifies tiny feature differences.

**Comparative Analysis for Similarity vs. Dissimilarity (RQ7 & RQ8)**. To investigate the relative efficacy of similarity versus dissimilarity in NSDA for medical image segmentation, we explore two alternative measures by rewriting Equation 3: a similarity measure ($a_{i,j} = \text{Sim}$) and a dissimilarity measure ($a_{i,j} = \text{EDiSim}(\cdot)$) different from the proposed Gaussian-kernel-based dissimilarity measure ($a_{i,j} = \text{DiSim}(\cdot)$). Specifically, the Gaussian kernel is directly used for the similarity measure, as it is the complementarity constraint for the proposed dissimilarity measure (Equation 3). While the Euclidean distance serves as a widely adopted dissimilarity metric, it inherently fails to satisfy two critical requirements of nonlinear transformation and $[0, 1]$ range constraints in attention mechanism design conventions. To address this limitation, we introduce a novel dissimilarity measure $\text{EDiSim}(\cdot)$ by applying a sigmoid function to the Euclidean distance, which is a standard method for producing

Table 4: The results (DSC (%)) of comparative analysis for similarity vs. dissimilarity in NSDA on the Synapse dataset. ♠, ♣, ◇, and ♡ denote U-Net, TransUNet, UNeXt, and TinyU-Net with NSDA, respectively. The highest score is marked in **bold**.

| Network | Rewriting Equation 3 | Mean | Spleen | Right Kidney | Left Kidney | Gallbladder | Liver | Stomach | Aorta | Pancreas |
|---|---|---|---|---|---|---|---|---|---|---|
| ♠ | $a_{i,j} = \mathrm{Sim}(\cdot)$ | 79.01 | 85.58 | 82.08 | 86.44 | 63.48 | 93.71 | 72.38 | **88.94** | 59.49 |
| | $a_{i,j} = \mathrm{EDiSim}(\cdot)$ | 79.51 | 87.02 | 84.25 | 87.05 | 63.06 | 93.49 | **73.15** | 88.24 | 59.82 |
| | Our original equation | **81.62** | **90.33** | **85.94** | **87.46** | **71.90** | **94.33** | 72.84 | 88.71 | **61.48** |
| ♣ | $a_{i,j} = \mathrm{Sim}(\cdot)$ | 81.14 | 86.26 | 83.77 | 89.36 | 70.94 | 95.20 | 72.39 | **89.57** | 61.64 |
| | $a_{i,j} = \mathrm{EDiSim}(\cdot)$ | 82.41 | 87.30 | 86.07 | 90.65 | **74.95** | 94.95 | 73.06 | 89.41 | 62.87 |
| | Our original equation | **83.81** | **93.04** | **87.83** | **91.52** | 73.36 | **95.52** | **76.31** | 89.23 | **63.66** |
| ◇ | $a_{i,j} = \mathrm{Sim}(\cdot)$ | 75.68 | 86.18 | 83.63 | 86.02 | 58.02 | 93.34 | **68.19** | 81.49 | 48.57 |
| | $a_{i,j} = \mathrm{EDiSim}(\cdot)$ | 75.72 | **88.62** | 82.53 | 86.40 | 52.22 | 92.88 | 67.14 | **83.08** | **52.91** |
| | Our original equation | **77.39** | 88.27 | **84.70** | **88.58** | **64.52** | **93.45** | 64.89 | 82.98 | 51.73 |
| ♡ | $a_{i,j} = \mathrm{Sim}(\cdot)$ | 80.15 | 90.68 | 85.56 | 89.96 | 64.48 | **93.96** | 70.76 | 87.43 | **58.35** |
| | $a_{i,j} = \mathrm{EDiSim}(\cdot)$ | 80.74 | **92.81** | **86.33** | 89.47 | 66.64 | 93.54 | 71.33 | 87.99 | 57.79 |
| | Our original equation | **81.57** | 92.16 | 86.21 | **91.29** | **71.16** | 93.82 | **72.13** | **88.81** | 56.95 |

attention weights [23]. In summary, the rewritten attention weight $a_{i,j}$ is expressed as follows:

$$\mathrm{Sim}(x_{i,j}, y_{i,j}, \sigma_{i,j}^2) = e^{\left(\frac{-\|x_{i,j} - y_{i,j}\|^2}{2\sigma_{i,j}^2}\right)}, \quad \mathrm{EDiSim}(x_{i,j}, y_{i,j}) = \mathrm{Sigmoid}(\|x_{i,j} - y_{i,j}\|), \quad (6)$$

where $\mathrm{Sim}(\cdot) \in [0, 1]$ denotes a Gaussian kernel for the similarity measure, and $\mathrm{EDiSim}(\cdot) \in [0, 1]$ denotes the proposed dissimilarity measure based on the sigmoid-activated Euclidean distance. As evidenced in Table 4, the dissimilarity-based operators ($\mathrm{EDiSim}(\cdot)$ and $\mathrm{DiSim}(\cdot)$) effectively enhance the segmentation accuracy compared to the similarity-based methods ($\mathrm{Sim}(\cdot)$) in the mean DSC (Table 1). Crucially, our Gaussian-kernel-based dissimilarity measure (Equation 3) achieves the most significant accuracy gains in neural networks for medical image segmentation. Although both $\mathrm{EDiSim}(\cdot)$ and $\mathrm{DiSim}(\cdot)$ are based on dissimilarity, they represent fundamentally different formulations: $\mathrm{EDiSim}(\cdot)$ relies on a simple geometric distance with fixed nonlinearity, while $\mathrm{DiSim}(\cdot)$ is derived from the complement of a Gaussian kernel, which inherently encodes a probabilistic notion of dissimilarity and aligns better with the statistical properties of feature distributions in medical images. Furthermore, we find that the proposed NSDA outperforms pairwise-similarity-based self-attention mechanisms (e.g., MHSA [25] and window-based MWSA [39]) in medical image segmentation (Table 1). These findings suggest a new insight that **medical image segmentation tasks benefit more from dissimilarity than similarity**, which aligns with clinical prior knowledge.

**Limitation**. Our method overcomes the accuracy-complexity trade-off paradox in attention mechanisms, enabling neural networks to achieve better performance gains. However, deploying attention-augmented neural networks in low-resource settings hinges not only on the computational complexity of attention modules, but also on the size of the network architecture. Consequently, lightweight neural networks are critical to advancing digital health equity, particularly in resource-limited settings. Future research will focus on their efficiency and scalability to bridge healthcare access divides.

## 6 Conclusion

Inspired by radiologists' neighborhood inspection and difference prioritization during image interpretation, we propose a parameter-free Neighborhood Self-Dissimilarity Attention in neural networks for medical image segmentation. Unlike traditional attention mechanisms, our method quantifies the element-neighborhood dissimilarity in feature maps to generate attention maps that prioritize salient regions with high disparity, thus enhancing segmentation accuracy without adding parameters directly related to computational complexity. This endeavor inspires clinical prior knowledge for the attention mechanism, offering a practical method to promote digital health equity in resource-limited settings.

## Acknowledgments and Disclosure of Funding

This work is partially supported by Sichuan Science and Technology Program (Nos. 2026NSF-SCZY0032, 2025ZNSFSC0735, and 2025ZNSFSC0509).

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
