# OpenReview forum: "Neighborhood Self-Dissimilarity Attention for Medical Image Segmentation"
_NeurIPS.cc/2025/Conference — NeurIPS 2025 spotlight_

### Official Review · Reviewer_WMJY · 2025-06-29

**Clarity:** 3
**Significance:** 3
**Originality:** 3
**Rating:** 4
**Confidence:** 3

**Summary:**

This paper proposes a novel, parameter-free attention mechanism called Neighborhood Self-Dissimilarity Attention (NSDA) for medical image segmentation. The core idea is inspired by radiologists' diagnostic process, where they focus on regions that exhibit high contrast or difference from their immediate surroundings. Instead of computing pairwise similarity like in conventional self-attention, NSDA calculates an attention score for each feature element based on its dissimilarity to the aggregated representation of its local neighborhood. The dissimilarity is quantified using a complement of a Gaussian kernel. The method also introduces a Dynamic Neighborhood Scaling (DyNS) strategy to adapt the size of the local neighborhood across different layers of the network, preventing feature homogenization. The authors claim this approach improves segmentation accuracy without adding computational complexity (in terms of parameters and MACs), thus addressing the performance-complexity trade-off and promoting health equity in resource-limited settings.

**Questions:**

1. The performance gains reported in Table 1 are impressive. Could the authors clarify if these results are from a single run or averaged over multiple runs? To strengthen the paper's claims, I would strongly recommend that the authors run the experiments for the main backbones (e.g., U-Net and TinyU-Net) with at least 3-5 different random seeds and report the mean and standard deviation for the Dice scores. This would provide crucial evidence of the statistical significance of the improvements over baselines. An update with this information would likely increase my overall score.

2. The DyNS strategy uses a fixed scale factor S=8. The paper would benefit from a discussion on how this value was chosen. Could the authors provide a brief sensitivity analysis showing how performance (e.g., mean DSC on Synapse) varies with different values of S? This would help to better understand the robustness of this heuristic.

3. The generalization experiments in Table 2 show that NSDA performs well on the natural image dataset VOCSeg. This suggests that "dissimilarity" might be a more general principle for attention, extending beyond its motivation as a "clinical prior". Could the authors elaborate on why they believe this principle generalizes so effectively outside of medical imaging, where the "lesion vs. background" intuition might be less directly applicable?

**Ethical Concerns:**

["NO or VERY MINOR ethics concerns only"]

**Final Justification:**

The author resolved the majority of the issues.

**Limitations:**

Yes. The authors have adequately addressed the limitations in a dedicated paragraph. They correctly point out that while their attention module is lightweight, the overall deployment feasibility still depends on the size of the backbone network architecture, and they suggest future work on lightweight networks. This is a fair and honest assessment.

**Paper Formatting Concerns:**

None.

**Quality:**

3

**Strengths And Weaknesses:**

Strengths:

1. The core idea of a parameter-free, dissimilarity-based attention mechanism is highly original and significant. It is well-motivated by clinical practice and directly addresses the critical performance-complexity trade-off, making it highly practical for resource-constrained applications.

2. The method is tested on multiple backbones and datasets against a wide array of baselines. Furthermore, strong ablation studies clearly demonstrate the contribution of each proposed component (DyNS and the dissimilarity measure).

3. The paper is very well-written, clearly structured, and easy to follow, which significantly enhances the accessibility of its contributions.

Weaknesses:

1. The performance gains reported in Table 1 are impressive. Could the authors clarify if these results are from a single run or averaged over multiple runs? To strengthen the paper's claims, I would strongly recommend that the authors run the experiments for the main backbones (e.g., U-Net and TinyU-Net) with at least 3-5 different random seeds and report the mean and standard deviation for the Dice scores. This would provide crucial evidence of the statistical significance of the improvements over baselines. An update with this information would likely increase my overall score.

2. The DyNS strategy uses a fixed scale factor S=8. The paper would benefit from a discussion on how this value was chosen. Could the authors provide a brief sensitivity analysis showing how performance (e.g., mean DSC on Synapse) varies with different values of S? This would help to better understand the robustness of this heuristic.

3. The generalization experiments in Table 2 show that NSDA performs well on the natural image dataset VOCSeg. This suggests that "dissimilarity" might be a more general principle for attention, extending beyond its motivation as a "clinical prior". Could the authors elaborate on why they believe this principle generalizes so effectively outside of medical imaging, where the "lesion vs. background" intuition might be less directly applicable?

4. The experiments integrate the attention module only at the terminal layer of the feature extractor. In U-Net like architectures, attention is often applied at multiple stages of the encoder and decoder. An investigation into how the performance of NSDA changes with different placement strategies would have provided deeper insights into its mechanism.

---

> ### Author Rebuttal · Authors · 2025-07-29
>
> We deeply appreciate your professional reviews and constructive suggestions. Your comments are excellent and thank you for your approval of our study. We are delighted by your praise for the originality and significance of our methodology. In addition, we are pleased that you have praised our paper for being well written, clearly structured and easy to understand. We answer the questions as follows.
>
> > W1/Q1: The performance gains reported in Table 1 are impressive.  Could the authors clarify if these results are from a single run or averaged over multiple runs?  To strengthen the paper's claims, I would strongly recommend that the authors run the experiments for the main backbones (e.g., U-Net and TinyU-Net) with at least 3-5 different random seeds and report the mean and standard deviation for the Dice scores.  This would provide crucial evidence of the statistical significance of the improvements over baselines.  An update with this information would likely increase my overall score.
>
> A1: We deeply appreciate your constructive suggestions to strengthen the paper's claims. **Table 1 reports peak performance from a single run**, consistent with common practice in publications like TMI, CVPR and NeurIPS.  We appreciate and value your insight. However, generating comprehensive mean and standard deviation significantly increases computational costs due to rerunning all experiments multiple times, which is challenging given our lab's GPU constraints and the rebuttal timeframe.  Nonetheless, **we have made every effort to supplement experiments and reported mean DSC (%)**: we prioritized five recent advanced attention methods and conducted experiments with three different random seeds on the challenging Synapses dataset using main backbones (U-Net and TinyU-Net).
>
> |Attention|Source|U-Net|TinyU-Net|
> |-|:-:|:-:|:-:|
> |||77.78$\pm$0.55|78.12$\pm$0.59|
> |+ CGA|TIP'24|77.11$\pm$0.89|79.20$\pm$0.77|
> |+ CMO|TPAMI'24|76.45$\pm$0.67|79.11$\pm$0.51|
> |+ MLKA|CVPR'24|78.71$\pm$0.53|80.09$\pm$0.66|
> |+ MSCAM|CVPR'24|75.92$\pm$1.12|44.59$\pm$6.35|
> |+ MWSA|AAAI'25|47.36$\pm$4.15|72.71$\pm$0.64|
> |+ NSDA|Ours|**81.35$\pm$0.52**|**81.09$\pm$0.46**|
>
> The updated results (avg ± std) consistently confirm our method's superiority. We sincerely appreciate your constructive feedback and the opportunity to enhance our work. We will continue to conduct experiments to provide statistically significant evidences/results.
>
> > W2/Q2: The DyNS strategy uses a fixed scale factor S=8.  The paper would benefit from a discussion on how this value was chosen.  Could the authors provide a brief sensitivity analysis showing how performance (e.g., mean DSC on Synapse) varies with different values of S?  This would help to better understand the robustness of this heuristic.
>
> A2: Thank you for this insightful suggestion. **We conduct a sensitivity analysis of scale factor $S$** on the Synapse dataset and report mean DSC (%).
>
> |Network|S=2|S=4|S=8 (Ours)|S=16|S=32|
> |-|-|-|-|-|-|
> |NSDA-augmented U-Net|80.68$\pm$0.80|79.96$\pm$0.55|**81.35$\pm$0.52**|79.74$\pm$0.64|78.94$\pm$0.83|
> |NSDA-augmented TinyU-Net|80.54$\pm$0.69|79.37$\pm$0.82|**81.09$\pm$0.46**|79.51$\pm$0.74|80.29$\pm$0.55|
>
> We find that the optimal performance at $S=8$. We hypothesize that smaller $S$ expands fields but risks diluting local features, while larger $S$ excessively constrains contextual capture. This trade-off explains the robustness of $S=8$.
>
> Thank you again for your careful review. We will optimize the description with respect to $S$ for better understand.
>
> > W3/Q3: The generalization experiments in Table 2 show that NSDA performs well on the natural image dataset VOCSeg.  This suggests that "dissimilarity" might be a more general principle for attention, extending beyond its motivation as a "clinical prior".  Could the authors elaborate on why they believe this principle generalizes so effectively outside of medical imaging, where the "lesion vs. background" intuition might be less directly applicable?
>
> A3: Thanks for your thoughtful observation regarding our generalization to natural image benchmark VOCSeg, which helps us clarify the broader applicability for our approach. While our method was initially inspired by the clinical prior of lesion-background dissimilarity in medical imaging, **the underlying "dissimilarity" principle extends beyond this specific context** because it captures a universal property of visual information: meaningful regions, whether lesions in medical images or objects in natural scenes, are inherently distinguishable from their surroundings through distinct feature patterns. In medical imaging, this manifests as the contrast between pathological and normal tissues, but in natural images like VOCSeg, it translates to differences in texture, color, or structure that define object boundaries or semantic entities. The clinical prior served as a concrete motivation, but the core mechanism relies on detecting such intrinsic feature discrepancies, a cue that remains critical for segmentation tasks across domains even when the "lesion vs. background" intuition is less direct.
>
> We appreciate your observation and will elaborate on this connection between the clinical inspiration and the principle’s generalizability in the revised manuscript.
>
> > W4: The experiments integrate the attention module only at the terminal layer of the feature extractor.  In U-Net like architectures, attention is often applied at multiple stages of the encoder and decoder.  An investigation into how the performance of NSDA changes with different placement strategies would have provided deeper insights into its mechanism.
>
> A4: Thank you for your thoughtful feedback. We clarify that **a feature extractor is deployed as an independent component at every stage of both the encoder and decoder**, and the attention is applied to each extractor's output. We will refine this description in the final version of the paper.
>
> While our primary focus is architectural innovation rather than placement strategy in the attention mechanism, we appreciate your perspective. Accordingly, **we conducted a brief ablation study regarding attention’s placement** using U-Net on the Synapse dataset.
>
> |NSDA-augmented Encoder|NSDA-augmented Decoder|mDSC (%)|
> |-|-|-|
> |||77.78$\pm$0.55|
> |$\checkmark$||79.56$\pm$0.78|
> ||$\checkmark$|79.23$\pm$0.46|
> |$\checkmark$|$\checkmark$|**81.35$\pm$0.52**|
>
> Results indicate that applying NSDA to both encoder and decoder stages outperforms using it solely in either the encoder or decoder, maximizing its effectiveness.

---

### Official Review · Reviewer_xBWV · 2025-06-30

**Clarity:** 4
**Significance:** 3
**Originality:** 3
**Rating:** 5
**Confidence:** 4

**Summary:**

Authors propose a novel Neighborhood Dissimilarity Attention mechanism to aid medical image segmentation in resource-limited settings. Based on the way radiologists inspect neighboring pixels or elements in an image, Authors propose a weighting of spatial features based on the level of dissimilarity between each pixel and those of its neighborhood. Dynamic Neighborhood Scaling is introduced to limit the neighborhood at subsequent model depths, to avoid feature homogenization. Attention Fusion is also used, whereby features are scaled by the attention weights and an added to the original features via a residual - this avoid additional complexity usually added with an MLP. Authors demonstrate their method in 3 distinct imaging domains - CT, MRI, and Ultrasound - as well as natural images. Results demonstrate that NDSA achieves most significant performance gains under test, and is able to perform well across architectures and domains. The paper shows that dissimilarity-based attention yields better performance than similarity based attention.

**Questions:**

In eq1, `S` is empirically set to 8 to help ensure the neighborhood of attention remains proportional to feature map scales across network depths. Could this be done more algorithmically using some aggregation of convolutional kernel size and stride etc.?

**Ethical Concerns:**

["NO or VERY MINOR ethics concerns only"]

**Final Justification:**

Based on Authors' rebuttal and discussion on these and other Reviewers' comments, I maintain the rating of 5: Accept.

**Limitations:**

Authors have not listed any limitations of their method in their Limitations section - instead, they have only noted that resource-limited setting need lower complexity approaches, which is what they address in their work. Authors should use this section to note actual limitations of their approach.

**Quality:**

3

**Strengths And Weaknesses:**

The authors set the context of their research well, and lay out a measured approach to understand how their proposed NDSA can enhance the performance of attention mechanisms in the low-resource setting.

In addition, Authors present a solid set of RQs, and demonstrate their approach on a diverse domains. Authors provide additional investigation by extending their analysis to the natural domain to assess generalization of their method. It is appreciated that Authors also compare against available resource-limited variants of attention mechanisms, given these are their main alternatives. Authors use appropriate metrics for analysis.

Authors have focused on the UNet architecture and its variants in this paper due to the resource-limited nature of their focus - it is known that alternative approaches such as transformer-based architectures can yield impressive results in image segmentation tasks. It would be beneficial for authors to compare their approach to other, more traditional attention mechanisms too.

---

> ### Author Rebuttal · Authors · 2025-07-29
>
> We deeply appreciate your thoughtful review, especially your endorsement of our writing clarity, research context, methodological rigor, solid experiments, targeted benchmarking, and cross-domain validation. This affirmation motivates our continued commitment to thorough research, and we address your questions below.
>
> > W1: Authors have focused on the UNet architecture and its variants in this paper due to the resource-limited nature of their focus - it is known that alternative approaches such as transformer-based architectures can yield impressive results in image segmentation tasks. It would be beneficial for authors to compare their approach to other, more traditional attention mechanisms too.
>
> A1: Thank you for your valuable suggestion. While our initial comparisons included transformer-based U-Net variants like TransUNet (Table 1), we appreciate your point about network architectures. In addition, the most classic attention mechanisms are SE and CBAM, which are widely used in various computational vision fields such as image classification, object detection, and semantic segmentation.  **We accept your suggestion to perform new experiments** with transformer-based SegFormer (NeurIPS’21) and the classic lightweight attention mechanism ULSAM on Synapse dataset.
>
> |Network|Params (M)|FLOPs (G)|Throughput (FPS)|mDSC (%)|
> |-|-|-|-|-|
> |SegFormer|3.716|3.399|134.59|68.91|
> |+ SE|3.728|3.399|129.26|69.92|
> |+ CBAM|3.729|3.401|134.59|69.88|
> |+ MSCAM|4.155|3.608|134.59|67.14|
> |+ ULSAM|3.719|3.403|134.59|69.43|
> |+ NSDA (Ours)|3.716|3.399|119.70|**71.40**|
>
> Results further demonstrate NSDA’s effectiveness even on transformer-based architectures.
>
> > Q1: In eq1, $S$ is empirically set to 8 to help ensure the neighborhood of attention remains proportional to feature map scales across network depths. Could this be done more algorithmically using some aggregation of convolutional kernel size and stride etc.?
>
> A2: Thank you for this insightful observation. You are absolutely right. DyNS is currently implemented by dividing the feature map size by $S$ (Equation 1). For computing statistics like an average $y_{i,j}$ in Equation 2, we use pooling kernel size of ($b_L\times b_L$) (Equation  1)  for efficient calculation.
>
> We appreciate your attention to this detail. **The PyTorch code will be released upon publication**.
>
> > L1: Authors have not listed any limitations of their method in their Limitations section - instead, they have only noted that resource-limited setting need lower complexity approaches, which is what they address in their work. Authors should use this section to note actual limitations of their approach.
>
> A3: Thank you very much for your insightful suggestion on the Limitations section. **We agree with you and illustrate the practical limitations of our approach**. Our method operates on the spatial dimensions of feature maps, without taking the channel dimension of feature maps into account. Recent studies have indicated that there is redundancy in the channels of feature maps. How to prioritize the channels that are important to network performance in NSDA using a parameter-free method will be our future work. This is expected to provide insights for the model's structure optimization and resource allocation.
>
> We will revise and improve our Limitations section accordingly. Thank you for your positive feedback and constructive suggestions on our paper.

---

> > ### Comment · Reviewer_xBWV · 2025-08-04
> > **Acknowledgement of Author Response**
> >
> > This Reviewer thanks the Authors for their considered response, not only to this Review, but to other Reviewers comments and suggestions. Reading Authors' response to other Reviewers' questions satisfies me that the Authors have tried to demonstrate utility with their approach. However, it is also clear that there are some flaws in the design and questions about novelty and effectiveness compared with other approaches. I yield to the other reviewers on that, and look forward to the discussion.

---

> > > ### Author Response · Authors · 2025-08-05
> > > **Response for Reviewer xBWV regarding Methodological Novelty and Effectiveness**
> > >
> > > We sincerely appreciate the reviewers' thoughtful engagement with our work and their constructive critiques regarding novelty. We welcome this opportunity to clarify key distinctions for further discussion:
> > >
> > > **1. Clarification on Distinction from prior works (SimAM/ULSAM)**
> > >
> > > We respectfully clarify that SimAM and ULSAM do **not** leverage local statistics or subspace reconstruction errors to generate attention weights as stated by Reviewer MPPP. ULSAM explicitly employs $1\times 1$ group convolutions to generate attention weights, introducing trainable parameters. Both SimAM and ULSAM generate attention weights based on *global feature statistics*. Crucially, **our method is the first** to:
> > >
> > > - Employ a novel similarity-dissimilarity complementarity principle to elegantly transform the Gaussian kernel into a **dissimilarity measure** (Equation 3).
> > > - Introduce a Dynamic Neighborhood Scaling (DyNS) strategy (Equation 1). Inspired by radiologists' Neighborhood Inspection, DyNS **adaptively regulates** the neighborhood size across network hierarchies. This prevents overly small neighborhood (focusing only within ROIs) and overly large neighborhood (causing feature homogenization), striking an optimal balance (i.e., $S=8$ for DyNS strategy) validated experimentally.
> > >
> > > In Table 1 of the manuscript, we quantitatively compare NSDA with SimAM. Additionally, as presented in our earlier response, NSDA integrated with SegFormer as the backbone outperforms ULSAM under the same configuration. These results demonstrate that NSDA achieves superior accuracy gains over both SimAM and ULSAM, confirming its effectiveness. We believe this clarifies the reviewers' misconceptions and highlights the innovativeness of our approach.
> > >
> > > **2. Clarification on Distinction from Laplacian of Gaussian (LoG)**
> > >
> > > We appreciate Reviewer XYh8 for mentioning the comparison between the Laplacian of Gaussian ($LoG(x,y) = -\frac{1}{\pi\sigma^4} \left[ 1 - \frac{x^2 + y^2}{2\sigma^2} \right] e^{-\frac{x^2 + y^2}{2\sigma^2}}$) and our NSDA's dissimilarity measure ($\mathrm{DiSim}(x,y)=1-e^{\left ( \frac{-\left \| x-y \right \|^{2} }{2\sigma^{2}}  \right ) }$). However, we emphasize fundamental **differences** between the two:
> > >
> > > - *Mathematical Form & Purpose*: The Laplacian of Gaussian (LoG) is the second derivative of a Gaussian function. It serves as an edge detection operator: first smoothing the image with a Gaussian filter to attenuate high-frequency noise, then applying the Laplacian operator to detect edge contours. Conversely, NSDA is the **complement of the Gaussian kernel** ($1 – GaussianKernel$) as an **element-neighborhood dissimilarity measure**. We designed NSDA explicitly as a dissimilarity metric based on the similarity-dissimilarity complementarity principle. Since Gaussian kernels measure similarity (higher values for more similar elements), its complement naturally measures dissimilarity, aligning perfectly with our goal of highlighting divergent regions. Qualitative results (Fig. 2) clearly demonstrate NSDA identifies salient **discrepancy regions, not edge contours**.
> > >
> > > - *Core Innovation*: To the best of our knowledge, NSDA is the **first method** to compute attention weights based on the **dissimilarity** between an element and its adaptively local neighborhood on the feature map. This contrasts sharply with predominant similarity-based mechanisms (e.g., self-attention and its variants). Inspired by radiologists prioritizing differences, this offers a **novel paradigm** for attention design, validated by our strong experimental results.
> > >
> > > - *Practical Impact*: Beyond performance, NSDA's **parameter-free design** directly addresses efficiency and accessibility concerns in low-resource settings. Unlike traditional attention mechanisms (which use convolutions/MLPs with sigmoid/softmax, prioritizing performance over low-resource accessibility), our parameter-free NSDA achieves a superior **accuracy-complexity trade-off**, enhancing potential for real-world clinical deployment and digital health equity.
> > >
> > >
> > > We take your comments most seriously and have greatly benefited from your positive feedback and constructive suggestions. We would like to take this opportunity to further clarify the novelty and effectiveness of our approach. Thank you again for your constructive feedback. We hope our response has adequately addressed your concerns regarding the novelty and effectiveness of our method compared with other approaches, and welcome ongoing discussion.
> > >
> > > Sincerely,
> > >
> > > The Authors

---

### Official Review · Reviewer_MPPP · 2025-07-01

**Clarity:** 3
**Significance:** 3
**Originality:** 2
**Rating:** 4
**Confidence:** 4

**Summary:**

This paper proposes a clinically inspired Neighborhood Self-Dissimilarity Attention (NSDA) mechanism, drawing on radiological diagnostic principles to address the long-standing performance–complexity trade-off in attention design through a parameter-free formulation. To further mitigate feature homogenization, the authors introduce a Dynamic Neighborhood Scaling (DyNS) strategy.

**Questions:**

(1) In Fig. 3, the effectiveness of DyNS is demonstrated only on the Synapse dataset. However, its performance under small-object or densely structured targets remains untested, limiting conclusions about the generalizability and robustness of the DyNS design.

(2) The neighborhood size in Equation (1) is heuristically determined using a fixed scaling factor (S = 8) and a compensatory constant (c), but the paper lacks sensitivity analysis to evaluate how this setting behaves across diverse network hierarchies and segmentation scenarios.

(3) Although NSDA is described as “parameter-free,” the implementation involves per-pixel computation of local mean, variance, and exponential transformations, which introduce non-trivial computational cost and implicit hyperparameters. The paper does not provide an analysis of the resulting resource consumption or inference latency.

(4) While Fig. 3 demonstrates the advantage of replacing static neighborhood sizes with DyNS, the paper does not report the performance of NSDA when DyNS is entirely removed, leaving the necessity of this component only partially evaluated.

**Ethical Concerns:**

["NO or VERY MINOR ethics concerns only"]

**Final Justification:**

The authors have provide rebuttal to address the previous concerns, and several concerns have been addressed. Thus, I raise the score.

**Limitations:**

See weaknesses and questions.

**Quality:**

3

**Strengths And Weaknesses:**

Strengths:
(1) This paper presents a parameter-free Neighborhood Self-Dissimilarity Attention (NSDA) framework that enables neural networks to focus on element-neighborhood differences, thereby boosting segmentation accuracy.
(2) This paper introduces a Dynamic Neighborhood Scaling (DyNS) strategy in NSDA, which adaptively regulates NSDA’s neighborhood size across network hierarchies to prevent feature homogenization.
(3) Experimental results show the effectiveness of the proposed model.

Weaknesses:
(1) While medical image segmentation typically relies on shape priors, multi-scale context, and boundary cues, NSDA instead emphasizes element-to-neighborhood dissimilarity. The authors do not sufficiently explain why such a “self-dissimilarity” mechanism offers better discriminative power than direct features, nor do they clarify its rationality and advantages in structured scenarios such as connected anatomical regions.
(2) Although the authors claim to present the first parameter-free attention mechanism based on element–neighborhood dissimilarity, similar ideas have appeared in prior works such as SimAM and ULSAM, which leverage local statistics (e.g., mean, variance) or subspace reconstruction errors to generate attention weights. The paper does not adequately distinguish NSDA from these approaches, either conceptually or theoretically.
(3) The dissimilarity computation is based on a fixed Gaussian kernel, rather than a learnable or adaptive metric. The authors do not provide justification for the selection or regulation of the variance term, which may lead to either over-smoothing or noise amplification. It also remains unclear why this fixed-kernel approach is preferable to alternative similarity or distance measures for segmentation.
(4) The multi-scale attention fusion in NSDA lacks detail on how weights across scales are determined or balanced. Moreover, the absence of ablation experiments regarding fusion strategies weakens the empirical support for this design component.

---

> ### Author Rebuttal · Authors · 2025-07-30
>
> We thank the reviewer for affirming the merits of our paper, particularly its clear presentation and effective methodology. We address your questions as follows.
>
> > W1: While medical image segmentation typically relies on shape priors, multi-scale context, and boundary cues, NSDA instead emphasizes element-to-neighborhood dissimilarity. The authors do not sufficiently explain why such a “self-dissimilarity” mechanism offers better discriminative power than direct features, nor do they clarify its rationality and advantages in structured scenarios such as connected anatomical regions.
>
> A1: We address reviewer's concerns by clarifying the rationality and advantages of our approach. NSDA can identify boundary information by leveraging element-neighborhood dissimilarity. It enhances segmentation completeness near edges for structures like the pancreas, right ventricle, and benign breast tumors by guiding U-Net to focus on local regions (Figure 2). This is because elements at boundaries exhibit greater dissimilarity to their surrounding neighborhoods than those in interior regions. Hence, **NSDA amplifies discriminative signals in local regions by prioritizing zones of high dissimilarity**. Further, the quantitative results (Tables 1 and 2) and qualitative results (Figure 2) demonstrate the effectiveness of our approach.
>
> > W2: Although the authors claim to present the first parameter-free attention mechanism based on element–neighborhood dissimilarity, similar ideas have appeared in prior works such as SimAM and ULSAM, which leverage local statistics (e.g., mean, variance) or subspace reconstruction errors to generate attention weights. The paper does not adequately distinguish NSDA from these approaches, either conceptually or theoretically.
>
> A2: We clarify the distinctive aspects of NSDA compared to existing methods (e.g., SimAM, ULSAM), thus addressing reviewer's misconception regarding its conceptual and theoretical foundations. ULSAM explicitly employs 1×1 group convolutions to generate attention weights, introducing trainable parameters. Both SimAM and ULSAM generate attention weights based on global feature statistics. Specifically, **Our NSDA has the three main innovations**.
> - *Adaptive Localization*: Per-layer adaptive adjustment of neighborhood size (Equation 1) to prevent preventing feature homogenization from excessive long-range aggregation.
> - *Novel Dissimilarity Measure*: First Gaussian-kernel-based metric leveraging local mean/variance (Equation 3) for element-neighborhood comparisons, providing theoretical justification for attention weighting.
> - *Radiologist-inspired Parameter-Free Paradigm*: Unlike the traditional paradigm (Convolution/MLP layers combined with Sigmoid/Softmax), NSDA introduces a parameter-free architecture, simulating radiologists' Neighborhood Inspection and Difference Prioritization, offering new design insights for attention mechanisms.
>
> > W3: The dissimilarity computation is based on a fixed Gaussian kernel, rather than a learnable or adaptive metric. The authors do not provide justification for the selection or regulation of the variance term, which may lead to either over-smoothing or noise amplification. It also remains unclear why this fixed-kernel approach is preferable to alternative similarity or distance measures for segmentation.
>
> A3: To address reviewer's concerns, we clarify: (i) the rationale for our fixed Gaussian kernel and variance selection, and (ii) NSDA's advantages over alternative similarity or distance measures for segmentation. **Fixed dissimilarity measure ensures consistent cross-sample comparability** by preserving classical ML principles (e.g., SVM/PCA kernels), whereas learnable measures undermine fairness due to input-dependent variations. Our variance term (Equation 3) aligns with classical machine learning method (e.g., SVM and PCA). **We conduct a sensitivity analysis for the variance term** on the Synapse dataset and report mean DSC (%).
>
> |Network|$\sigma_{i,j}^2$|$2\sigma_{i,j}^2$ (Ours)|$4\sigma_{i,j}^2$|$6\sigma_{i,j}^2$|
> |-|-|-|-|-|
> |NSDA-augmented U-Net|80.52|**81.62**|79.33|79.05|
> |NSDA-augmented TinyU-Net|80.34|**81.57**|80.06|80.18|
>
> Results show our setting achieves optimal segmentation accuracy. This can be attributed that a larger variance term reduces sensitivity to feature differences, whereas a smaller one amplifies it.
>
> NSDA outperforms other similarity or distance metrics (Table 3) due to:
> - *Clinical relevance*: Lesions/organs exhibit high dissimilarity from surroundings, aligning with radiologists' focus on discrepancy regions;
> - *Technical superiority*: Unlike alternatives such as Euclidean distance requiring sigmoid/softmax to generate attention weights, our Gaussian-based measure provides inherent nonlinearity while eliminating sign-induced bias (Figure 1), ensuring fair contribution from opposing-polarity features.
>
> We will enhance the paper's corresponding descriptions to improve clarity
>
> > W4: The multi-scale attention fusion in NSDA lacks detail on how weights across scales are determined or balanced. Moreover, the absence of ablation experiments regarding fusion strategies weakens the empirical support for this design component.
>
> A4: The reviewer's concerns stem from a misunderstanding of our work. We clarify that **our NSDA does not incorporate multi-scale attention fusion**. Nevertheless, reviewer's insight has prompted plans to develop a multi-scale variant in future work.
>
> > Q1: In Fig. 3, the effectiveness of DyNS is demonstrated only on the Synapse dataset. However, its performance under small-object or densely structured targets remains untested, limiting conclusions about the generalizability and robustness of the DyNS design.
>
> A5: To solve reviewer's concerns, we clarify that the Synapse benchmark (widely adopted in top venues like CVPR/NeurIPS) inherently includes small objects (e.g., aorta/pancreas, Figure 2). **We additionally validated performance on the COVID-19 CT dataset (Cell, 2020)**, reporting DSC (%) for clinically significant small consolidations. This dataset is from ''Clinically Applicable AI System for Accurate Diagnosis, Quantitative Measurements, and Prognosis of COVID-19 Pneumonia Using Computed Tomography''
>
> |Network|Small Organ/Lesion|DyNS (Ours)|K=(03,03)|K=(07,07)|K=(15,15)|K=(31,31)|K=(41,41)|K=(51,51)|K=(H,W)|
> |-|-|-|-|-|-|-|-|-|-|
> |NSDA-augmented TransUNet|Pancreas|**63.66**|63.09|63.65|62.87|62.55|61.22|61.31|61.33|
> |NSDA-augmented TinyU-Net|Aorta|**88.81**|87.07|87.43|87.99|88.68|88.06|87.91|87.74|
> |NSDA-augmented UNet|Consolidation|**79.74**|79.38|79.13|79.33|79.42|79.40|78.52|78.18|
>
> Results demonstrate DyNS’s robustness across small and dense anatomical structures.
>
> > Q2: The neighborhood size in Equation (1) is heuristically determined using a fixed scaling factor (S = 8) and a compensatory constant (c), but the paper lacks sensitivity analysis to evaluate how this setting behaves across diverse network hierarchies and segmentation scenarios.
>
> A6: To solve reviewer's concerns, we clarify that the compensatory constant $c \in {0,1}$ is used for ensuring element’s bilateral symmetry within neighborhoods. **We conducted sensitivity analysis** for scaling factor $S$ on Synapse dataset , reporting mean DSC (%).
>
> |Network|S=2|S=4|S=8 (Ours)|S=16|S=32|
> |-|-|-|-|-|-|
> |NSDA-augmented U-Net|80.68$\pm$0.80|79.96$\pm$0.55|**81.35$\pm$0.52**|79.74$\pm$0.64|78.94$\pm$0.83|
> |NSDA-augmented TinyU-Net|80.54$\pm$0.69|79.37$\pm$0.82|**81.09$\pm$0.46**|79.51$\pm$0.74|80.29$\pm$0.55|
>
> Optimal performance occurs at $S=8$, as smaller $S$ expands receptive fields but diluting local features, while larger $S$ limits contextual capture.
>
> > Q3: Although NSDA is described as “parameter-free,” the implementation involves per-pixel computation of local mean, variance, and exponential transformations, which introduce non-trivial computational cost and implicit hyperparameters. The paper does not provide an analysis of the resulting resource consumption or inference latency.
>
> A7: To solve reviewer's concerns, we clarify that our "parameter-free" claim refers exclusively to the absence of learnable parameters (e.g., convolution/MLP weights) that increase model size and require gradient updates. **While using statistical operations (e.g., mean/variance), these leverage highly optimized PyTorch tensor primitives with substantially lower cost** than parametric operations (e.g., convolutions/matrix multiplications), which are the the primary FLOPs/MACs contributors in deep networks. Thus, the pytorch implementation of NSDA is efficient and lightweight. The code is available upon publication.
>
> As shown in Table 1, NSDA maintains near-identical parameter counts and FLOPs/MACs versus base networks (e.g., 19.58M params in NSDA-augmented U-Net matching original U-Net), while achieving higher throughput than most baselines (e.g., 105.37 FPS for NSDA-augmented U-Net vs. 82.78 FPS for CBAM-augmented U-Net). Results confirm NSDA introduces negligible computational overhead and causes no significant inference delays.
>
> We will implement the suggested analyses and clarifications during revision to strengthen claims.
>
> > Q4: While Fig. 3 demonstrates the advantage of replacing static neighborhood sizes with DyNS, the paper does not report the performance of NSDA when DyNS is entirely removed, leaving the necessity of this component only partially evaluated.
>
> A8: The reviewers' concerns mainly stem from misunderstandings of our work. Complete removal of DyNS would render neighborhood size $b_L$ (Equation 1) undefined, necessitating manual setting of a static size $K$. Only when a static neighborhood size $K$ is established can the element-neighborhood dissimilarity measure (Equation 2) remain mathematically defined; otherwise, **removal of DyNS renders this measure inoperable**.
>
> We will clarify this architectural necessity in the paper to prevent misunderstandings and enhance overall clarity.

---

### Official Review · Reviewer_XYh8 · 2025-07-03

**Clarity:** 3
**Significance:** 2
**Originality:** 3
**Rating:** 4
**Confidence:** 4

**Summary:**

The authors propose a different, parameter-free, attention mechanism, NSDA, that is more efficient than pairwise-similarity based mechanisms and shows promisiong results for segmentation tasks. As part of NSDA, DyNS is proposed to constrain NSDA to localized information throughout feature hierarchies present in deeper networks. The authors emphasize that both, NSDA and DyNS, are inspired by patterns present in how actual radiologists' workflows. The paper has strong experiments to demonstrate the efficacy of the method and support parameter choices.
However, the major shortcoming of the paper is that it is poorly motivated in that a more efficient attention mechanism for medical image segmentation tasks doesn't actually address any of the health equity concerns the authors claim to care about.

**Questions:**

N/A

**Ethical Concerns:**

["NO or VERY MINOR ethics concerns only"]

**Final Justification:**

The authors have addressed my concerns.

**Limitations:**

Yes.

**Paper Formatting Concerns:**

N/A.

**Quality:**

3

**Strengths And Weaknesses:**

Strengths:
- The experiments are thorough and inspire confidence in the method's efficacy.

Weaknesses:
- The clinical motivation for this paper the authors present seems to be poorly suited for the task and model presented. What would the segmentation of lesions be needed for in a resource-scarce location? Who is taking the images and who is making the decision to biopsy after the segmentation is completed? The authors seem to think a focus on regions with scarce resources and health equity will get their paper more traction, but have not done any of the legwork necessary to ensure their method is applicable, or even interesting, for these situations. This weak motivation distracts from the main contribution of the paper and frankly, makes little sense. The brain MRI images in Figure 1 don't seem to represent a modality which would be available in a resource-scarce location.
- The authors state that (paraphrased): regions with small ROIs are ignored by existing medical image segmentation methods. This is demonstrably untrue. See https://pmc.ncbi.nlm.nih.gov/articles/PMC9690845/ and https://link.springer.com/article/10.1007/s11432-021-3340-y and https://link.springer.com/article/10.1007/s12652-021-03358-8 for a few examples in breast imaging.
- The authors Neighborhood Inspection and Difference Prioritization modules seem to be re-deriving traditional radiomic features. Localized variance and mean has been used to describe medical imaging texture for decades. Equation 3 seems to be conceptually similar to a Laplace of a Gaussian feature.
- BUSI is a low-quality dataset in which about 30% of images contain measurement calipers, providing the model an ROI from which to segment. The authors would be better served testing their method on a cleaner dataset. See Figure 2 for an example lesion with measurement calipers which much be placed by an expert reader such as a sonographer. Which would be unavailable in the authors application area.
- Again, CT and MRI are not going to be available in a resource-scarce locale. The authors contradict their motivation once again.

---

> ### Author Rebuttal · Authors · 2025-07-31
>
> We appreciate the reviewer's positive feedback on our solid experiments and the demonstrated effectiveness of NSDA in medical image segmentation tasks.
>
> Regarding the reviewer's concern about our motivation, we respectfully clarify that our focus on digital health equity arises from the inability of lightweight architectures in resource-limited settings to integrate advanced attention mechanisms. These mechanisms currently face an accuracy-complexity trade-off paradox: accuracy gains demand higher computational costs, while reducing complexity sacrifices model accuracy. This limitation prevents equitable access to high-accuracy attention mechanisms in constrained settings, exacerbating digital health divides. By facilitating deployment on edge/mobile/ wearable/portable devices, **our parameter-free NSDA expands accessibility of expert-level digital segmentation in low-resource settings**. Our study aligns with the World Health Organization's **vision for universal health coverage**, responding to one of its core requirements: ensuring equitable access to high-performance digital health technologies within low-resource settings (WHO: Global Strategy on Digital Health 2020-2025).
>
> We responded to the reviewer's questions point by point.
>
> > W1: The clinical motivation for this paper the authors present seems to be poorly suited for the task and model presented. (i) What would the segmentation of lesions be needed for in a resource-scarce location? (ii) Who is taking the images and who is making the decision to biopsy after the segmentation is completed?  (iii) The authors seem to think a focus on regions with scarce resources and health equity will get their paper more traction, but have not done any of the legwork necessary to ensure their method is applicable, or even interesting, for these situations. This weak motivation distracts from the main contribution of the paper and frankly, makes little sense. (iv) The brain MRI images in Figure 1 don't seem to represent a modality which would be available in a resource-scarce location.
>
> A1: We appreciate the reviewer's critical perspective on our motivation. The reviewers' concerns arose primarily from a misunderstanding of our study's motivation. To clarify, **we mainly focus on low-resource settings/devices rather than resource-scarce location**.
>
> (i) We appreciate the reviewer raising this question regarding the clinical necessity of lesion segmentation in resource-scarce location. Lesion segmentation plays a critical role even in such contexts by providing qualitative and quantitative evidence about lesion size, shape, and location, which supports disease diagnosis and informs radiomic analyses. This information is vital for applications like surgical navigation, evidence-based medicine, and personalized diagnosis. In resource-scarce location, where expert radiologists are scarce, **AI-assisted lesion segmentation can empower primary care providers to better identify and assess anatomical structures and abnormalities in medical images**, helping bridge the gap in diagnostic capacity. Therefore, our work addresses a fundamental need in these underserved areas by making advanced diagnostic capabilities more accessible.
>
> (ii) We appreciate the reviewer's pertinent questions regarding the process of image acquisition and subsequent biopsy decisions. We clarify that **our work focuses exclusively on computational segmentation of medical images**.  Hence, we assume availability of pre-acquired images by low-resource settings like Point-of-Care (POC) Imaging, while **biopsy decisions is a downstream clinical task**. Our segmentation provides quantitative lesion measurements (e.g., shape/size) that assist clinicians in biopsy decisions.
>
> (iii) We appreciate the reviewer's feedback regarding motivation. Our core intention was to establish a technical pathway inspired by radiologists' diagnostic patterns: NSDA enables neural networks to achieve performance gains without computational overhead by modeling local element neighborhood differences through parameter-free design. Critically, **our method enables integration into lightweight architectures, making high-accuracy segmentation feasible in low-resource settings/devices** like portable devices (e.g., Butterfly iQ+) or wearable monitors for point-of-care use in homes, field clinics, or emergencies [1][2][3].
>
> (iv) We appreciate the reviewer's careful observation. We clarify that the images in Figure 2 are cardiac cine-MRI sequences from the ACDC benchmark, not brain MRI. We acknowledge that traditional high-field MRI systems remain limited in resource-scarce regions. We mainly focused on low-resource settings/devices rather than resource-scarce location. **Emerging low-resource settings such as portable and low-field MRI systems (e.g., Hyperfine's AI-powered portable MRI [4]) can take MRI**. This technological progress supports for experiments using MRI data.
>
> > W2: The authors state that (paraphrased): regions with small ROIs are ignored by existing medical image segmentation methods. This is demonstrably untrue. See *ESTAN: Enhanced Small Tumor-Aware Network for Breast Ultrasound Image Segmentation. Healthcare (Basel), 2022.*, *Difficulty-aware prior-guided hierarchical network for adaptive segmentation of breast tumors. Sci. China Inf. Sci., 2023.* and *Auxiliary diagnosis of small tumor in mammography based on deep learning. J Ambient Intell Human Comput, 2023.* for a few examples in breast imaging.
>
> A2: We appreciate the reviewer's careful attention, which has helped us identify an imprecision in our wording. The reviewer's concerns stem from the description of a sentence in the Related Work section. This description is in the original text: "*current methods still process all regions of an image equally during feature learning, failing to pay attention to diagnostically vital ROIs*". This was intended to refer to the specific methods discussed in the preceding context, not all existing methods broadly. To clarify, **we will revise "current methods" to "the above methods" in the revised manuscript**, ensuring the statement accurately reflects our focus on the limitations of the approaches cited earlier.
>
> > W3: The authors Neighborhood Inspection and Difference Prioritization modules seem to be re-deriving traditional radiomic features. Localized variance and mean has been used to describe medical imaging texture for decades. Equation 3 seems to be conceptually similar to a Laplace of a Gaussian feature.
>
> A3: We appreciate the reviewer's astute observation linking our work to traditional radiomic features, which provides a valuable opportunity to clarify novelty. While localized mean and variance have long been used in radiomics for handcrafted texture analysis, our Neighborhood Inspection and Difference Prioritization modules integrate these statistics dynamically within a neural network's feature learning process—adapting to hierarchical feature maps and enabling end-to-end optimization, rather than serving as offline, precomputed features. Equation 3, though leveraging mean and variance, is designed to quantify element-neighborhood dissimilarity via a Gaussian kernel complement, guiding neural networks to focus on local salient region. This stands apart from Laplacian of Gaussian features, which focus on edge detection in raw images. **Our method simulates radiologists' Neighborhood Inspection and Difference Prioritization to quantify element-neighborhood discrepancies**, enabling neural networks to focus on salient regions crucial to segmentation.
>
> We will emphasize these distinctions in the revised manuscript to better highlight the novel integration of these concepts within attention mechanisms.
>
> > W4: BUSI is a low-quality dataset in which about 30% of images contain measurement calipers, providing the model an ROI from which to segment. The authors would be better served testing their method on a cleaner dataset. See Figure 2 for an example lesion with measurement calipers which much be placed by an expert reader such as a sonographer. Which would be unavailable in the authors application area.
>
> A4: We appreciate the reviewer's careful note on the BUSI dataset. We clarify that **measurement calipers occlude tumor regions rather than providing segmentation cues**. As shown in Figure 2, they introduce occlusions that complicate segmentation. Since real-world ultrasound often contains noise (including calipers), BUSI remains a clinically relevant benchmark validated in TMI/CVPR/NPJ Digit. Med. Notably, U-Net struggles with such noise, while NSDA improves accuracy by focusing on genuine lesion discrepancies (Figure 2). As recommended by the reviewer, **we have added experiments on the high-quality TNSCUI2020 ultrasound dataset**.
>
> |Network|mDSC (%)|
> |-|-|
> |TinyU-Net|84.60|
> |+ SE|84.87|
> |+ CBAM|85.02|
> |+ MSCAM|83.96|
> |+ MLKA|84.35|
> |+ NSDA (Ours)|**85.89**|
>
> Results confirm NSDA's effectiveness there as well.
>
> > W5: Again, CT and MRI are not going to be available in a resource-scarce locale. The authors contradict their motivation once again.
>
> A5: We appreciate the reviewer's concern. It's worth noting that recent advancements in Point-of-Care (POC) Imaging have yielded compact, low-resource devices, such as Hyperfine's POC MRI and Samsung's POC CT systems. **These POC devices are specialized for taking CT/MRI in resource-constrained settings**. Our method enables high-accuracy segmentation on resource-constrained emerging devices/settings via integration into neural networks, thereby advancing digital health equity.
>
> ---
>
> Reference:
>
> [1] "Unext: Mlp-based rapid medical image segmentation network." MICCAI, 2022.
>
> [2] "Overcoming barriers in the use of artificial intelligence in point of care ultrasound." npj Digital Medicine, 2024.
>
> [3] "See how your body works in real time—wearable ultrasound is on its way." Nature, 2024.
>
> [4] "Low-field MRI: clinical promise and challenges." J Magn Reson Imaging, 2023.

---

> > ### Comment · Reviewer_XYh8 · 2025-08-05
> >
> > Thank you for the thorough response and clarification. The clarification has been helpful in understanding the specific motivation of the paper. "Resource-scarce" still subjectively feels more broad than simply referring to regions with missing radiological expertise, which is a more narrow focus, but I will accept that it is clarified in the text.
> >
> > I appreciate the additional clarifications added to the text and especially the additional experiment added.
> > As such, I will raise my score.

---

> > > ### Author Response · Authors · 2025-08-06
> > > **Response for Reviewer XYh8**
> > >
> > > Thank you for your careful review and the opportunity to clarify the motivation for the paper.  We sincerely appreciate your recognition of our work and your constructive feedback, especially your acknowledgment of the value added by the additional experimentation.

---

### Note · Authors · 2025-08-11

Dear Reviewers, PCs, ACs, and SACs,

We sincerely appreciate your valuable insights and constructive feedback on our work. Reviewers consistently highlighted the following strengths:
- **Significance for Digital Health Equity**. Our parameter-free NSDA enables deployment on edge/mobile devices in low-resource settings, overcoming accuracy-complexity trade-off paradox, thus promoting the WHO’s vision for universal health coverage by democratizing expert-level medical segmentation.
- **Clinically Grounded Motivation**.  NSDA uniquely simulates radiologists’ diagnostic workflow: Neighborhood Inspection and Difference Prioritization to identify salient regions critical to medical imaging.
- **Original Contribution**. NSDA's novel dissimilarity measure quantifies element-neighborhood differences via Gaussian kernel complementarity. This mechanism assigns higher attention weights to regions with larger differences, guiding the neural network to focus on clinically salient regions and thereby enhancing its segmentation accuracy.
- **Rigorous Validation**. Extensive tests across multiple backbones/datasets, ablation studies, cross-task validation, and natural-image generalization demonstrate NSDA’s efficacy and robustness.

We thank Reviewers MPPP and xBWV for engaging with our novelty claims.  Our method delivers distinct conceptual and theoretical advances:
- **Adaptive Perception Region**: The proposed DyNS strategy adaptively adjusts neighborhood size per layer, preventing feature homogenization from excessive long-range aggregation.
- **Novel Dissimilarity Measure**: We propose the first dissimilarity measure based on the complement of a Gaussian kernel which exploits the similarity-dissimilarity complementarity principle to quantify element-neighborhood differences.
- **Radiologist-inspired Parameter-Free Paradigm**: Unlike traditional paradigms using Convolution/MLP layers and Sigmoid/Softmax, NSDA's parameter-free architecture simulates radiologists' Neighborhood Inspection and Difference Prioritization, providing a novel attention mechanism design principle.

We thank reviewers for their valuable feedback, which was addressed point-by-point in our rebuttal. The manuscript has been revised for clarity based on these insights.

We are encouraged by reviewers’ recognition of NSDA's potential to inspire the **Digital Health Equity** community and believe this work merits publication to stimulate further discussion.

Thank you for your time and consideration.

---

### Decision · Program_Chairs · 2025-09-17

**Decision:**

Accept (spotlight)

**Comment:**

This paper presents a novel Neighborhood Self-Dissimilarity Attention for medical image segmentation. The authors validate the effectiveness and generality of the proposed attention with four segmentation baseline, including Unet, TransUnet, UNeXt, and TinyU-Net. The experimental results consistently show the advantages of the NSDA module. Overall, the reviewers are supportive to this paper.